# Epigenetic Alterations in Hepatocellular Carcinoma: Mechanisms, Biomarkers, and Therapeutic Implications

**DOI:** 10.3390/ph18091281

**Published:** 2025-08-27

**Authors:** Adil Farooq Wali, Abid Reza Ansari, Prince Ahad Mir, Mohamed El-Tanani, Rasha Babiker, Md Sadique Hussain, Jasreen Uppal, Asma Ishrat Zargar, Reyaz Hassan Mir

**Affiliations:** 1RAK College of Pharmacy, RAK Medical and Health Sciences University, Ras Al Khaimah P.O. Box 11172, United Arab Emirates; 2Laboratory of Epigenetics and Diseases, Department of Pharmacology and Toxicology, National Institute of Pharmaceutical Education and Research, S.A.S. Nagar (Mohali) 160062, Punjab, India; 3Department of Pharmacognsoy and Phytochemistry, Khalsa College of Pharmacy, G.T. Road, Amritsar 143002, Punjab, India; 4Department of Physiology, RAK College of Medical Sciences, Ras Al Khaimah Medical and Health Sciences University, Ras Al Khaimah P.O. Box 11172, United Arab Emirates; 5Uttaranchal Institute of Pharmaceutical Sciences, Uttaranchal University, Dehradun 248007, Uttarakhand, India; sadiquehussain007@gmail.com; 6Pharmacology Division, Department of Pharmaceutical Sciences, University of Kashmir, Hazratbal, Srinagar 190006, Jammu and Kashmir, India; 7Pharmaceutical Chemistry Division, Department of Pharmaceutical Sciences, University of Kashmir, Hazratbal, Srinagar 190006, Jammu and Kashmir, India

**Keywords:** DNA methylation, epigenetics, histone methylation, liver damage, ncRNAs (noncoding ribonucleic acid)

## Abstract

Hepatocellular carcinoma (HCC), the most prevalent primary liver cancer, continues to pose a significant global health burden due to its high mortality rate. In addition to genetic alterations, epigenetic aberrations, including DNA methylation, histone modifications, chromatin remodeling, and noncoding RNA (ncRNA) dysregulation, play critical roles in HCC initiation and progression. Notably, miR-375 and miR-483-5p are among the most dysregulated miRNAs in HCC, with their altered expression levels closely associated with tumor stage and patient survival. These epigenetic modifications offer promising therapeutic avenues due to their reversibility and dynamic nature. Furthermore, specific epigenetic signatures such as CDH1 promoter hypermethylation and HOTAIR overexpression are being explored as potential biomarkers for early detection and treatment response. In this chapter, we review recent advances in the epigenetic landscape of HCC and discuss their diagnostic and therapeutic implications, highlighting their potential to improve patient outcomes through personalized medicine approaches.

## 1. Introduction

Hepatocellular carcinoma (HCC), the most common form of primary liver cancer, is a highly lethal malignancy with a complex molecular etiology involving both genetic and epigenetic alterations. Globally, liver cancer accounts for over 800,000 deaths annually, ranking as the third leading cause of cancer-related mortality, and second leading cause of cancer-related mortality in men [1]. HCC represents approximately 90% of all primary liver cancer cases, with rising incidence driven by metabolic syndrome, viral hepatitis, and alcohol-related liver disease [2,3]. The transition of a normal cell to a cancer cell is driven by the alterations in the DNA of a cell, also known as mutations. Proto-oncogenes (Ras, HER2, Myc, cyclin D) and tumor suppressor genes (TSGs: BRCA1, BRCA2, p53, TP53) regulate normal cell division; the former cause cells to multiply, and the latter stop cells from multiplying, respectively [4,5]. Mutations in these genes lead to uncontrolled cell division and the formation of cancer [6]. Risk factors vary depending on the cancer type and may include exposure to carcinogens, chronic inflammation, genetic predisposition, age, lifestyle, and environmental factors [7]. While genetic mutations in oncogenes and TSGs such as *TP53* and *CTNNB1* are well-established in HCC [8], recent advances underscore the pivotal role of epigenetic dysregulation in liver carcinogenesis [9]. Epigenetics refers to all stable variations in phenotypic features that are not encoded in the DNA sequence itself [10,11]. Epigenetic mechanisms retain heritable modifications in gene expression and chromatin organization over numerous cell generations [12]. Cancer cells have the property to metastasize, i.e., they spread to other parts of the body from their place of origin. Cancer is the second-highest death-causing disease after ischemic heart disease (IHD). Among all cancer types, lung, stomach, and liver are the deadliest ones in the general population [13,14,15]. The epigenetic landscape of HCC involves multiple layers of regulation, all of which contribute to tumor progression, metastasis, and therapy resistance. These mechanisms and their associated biomarkers are summarized in Table 1.

HCC is mainly caused by cirrhosis of any etiology [16]; however, in chronic hepatitis B virus (HBV) infection, approximately 15–20% of patients may develop HCC in the absence of cirrhosis, owing to the direct oncogenic potential of the virus and integration of HBV DNA into the host genome [17]. Due to an increase in risk factors like diabetes and obesity, HCC incidence rates are also rising. Other well-known risk factors that lead to disturbances in hepatic epigenome include chronic hepatitis B and C, alcohol induced cirrhosis, hemochromatosis, exposure to dietary aflatoxin, steatohepatitis, and smoking [18]. Over the past decades, HCC has been viewed as a long-term progressive multistep process driven by the genetic mutations in oncogenes and TSGs, p53 and β-catenin being the major ones [19]. However, a recent uptick in research highlighting the importance of epigenetic mechanisms has significantly altered the perception of hepatocellular carcinoma (HCC) as a genetic disease only [20,21]. The hepatic epigenome is uniquely responsive to environmental stressors, including viral infections, metabolic dysfunction, and xenobiotic exposure [19], which can lead to persistent changes in chromatin structure and gene expression relevant to HCC initiation and progression [22]. Key epigenetic alterations in HCC include global DNA hypomethylation, promoter hypermethylation of TSGs, histone modifications, and altered expression of non-coding RNAs such as miR-122, H19, and various lncRNAs. Overexpression of DNA methyltransferase 1 (DNMT1) and enhancer of zeste homolog 2 (EZH2) has also been frequently observed in HCC and correlates with poor prognosis [23] cause oncogenic transformations, destabilize the genome, change gene expressions, which cause uncontrolled cell proliferation and thus trigger hepatocellular carcinoma [24]. A combination of both the genetic and epigenetic factors forms the molecular basis of HCC [25]. Epimutations, defined as heritable aberrant changes in gene expression caused by epigenetic modifications rather than changes in DNA sequence, may occur at a higher rate compared to gene mutations and have a significant effect on HCC metastasis [26,27]. The development of next-generation DNA sequencing has greatly improved our understanding of the genetic and molecular pathobiology of hepatocellular carcinoma. Some well-known public cancer genomic databases like the International Cancer Genome Consortium (ICGC), the Cancer Genome Atlas (TCGA), and the International Human Epigenome Consortium-2010 (IHEC) have been launched to generate reference maps of human epigenomes and provide comprehensive knowledge about genomic abnormalities, key cellular stemness, cell differentiation, and proliferation. With the help of these multiple public databases, the understanding of epigenomic mechanisms implicated in HCC may greatly improve its prevention and treatment.

## 2. Epigenetics

Epigenetics represents one of the fastest and most rapidly changing branches of biological study nowadays. The findings of the epigenome have called into question the idea that the genetic code is the only foundation for biological heredity. It is now understood that DNA bases may be changed without changing the nucleotide sequence, and it is believed this “epigenome” can be controlled by a range of environmental variables such as chemicals, diet, surroundings, anxiety, and aging [28,29].

Conrad Waddington used the word epigenetics at the beginning of the 1940s to describe the “branch of biology which investigates the underlying relationships among genes and the outcomes that brought the phenotype into existence” [30]. Waddington’s description of epigenetics pertained to its role in embryonic growth, controlling variations in gene activity and inactivity that were linked to cell division. Subsequently, the definition of epigenetics has been revised to incorporate new molecular discoveries. It is currently more generally used to describe heritable shifts in gene activity that do not result from changing DNA sequences, mainly chromatin modifications such as DNA methylation, histone modification, and nucleosome placement [31,32,33,34]. However, the meaning of epigenetics has expanded throughout time, owing mostly to technical breakthroughs in the discipline. Epigenetics is now defined as the “learning of mitotically and/or meiotically heritable variations in gene functioning that cannot be attributed to differences in DNA sequence” [35,36]. DNA methylation, histone modification, and RNA-based mechanisms, i.e., non-coding RNAs (ncRNA) such as long non-coding RNAs (lncRNAs) and short non-coding RNAs (sncRNAs) are all established epigenetic procedures [37].

### 2.1. Epigenetic Modifiers

All biological processes that modify gene expression in a reversible, transmissible, and adaptable manner without changing the DNA sequence are included in epigenetic alterations. Epigenetic changes, as defined by Conrad Waddington, were first identified in the differentiation of cells [38]. Epigenetic modifiers are chemicals that influence gene expression by changing the structure and functionality of DNA and its linked proteins [39]. In contrast with disruptions in DNA sequence, epigenetic alterations are reversible and may be impacted by a variety of external variables such as nutrition, lifestyle, and toxic contact [40,41]. These changes can have a significant impact on cellular functions, growth, and vulnerability to diseases [42].

DNA methylation, which includes the incorporation of a methyl group into the DNA molecule and relates to gene silencing, represents one of the most well-studied epigenetic changes [43,44]. Another important part of epigenetic modulation is histone changes. Histones are proteins that wrap DNA into chromatin, resulting in a compact form [45]. Post-translational histone modifications such as acetylation, methylation, phosphorylation, and ubiquitination can influence chromatin structure, thereby modulating DNA accessibility and regulating gene expression [46,47]. Histone modifiers, which include histone acetyltransferases (HATs) and histone deacetylases (HDACs), can add or eliminate certain modifications [48,49]. Non-coding RNAs, including sncRNAs and lncRNAs, also play a role in epigenetic control. These RNAs can connect to DNA or RNA regions and influence gene expression by interfering with the transcription or translation mechanisms [50,51].

Because of their potential as medicinal targets for a variety of disorders, epigenetic modulators have received a lot of interest in the field of biomedical science. The epigenetic imbalance has been linked to a variety of illnesses, notably tumors [52,53], neurological disorders [54,55,56], heart disease [57,58,59], and developmental abnormalities [60,61,62]. As a result, scientists are currently investigating the application of inhibitors of small molecules and stimulants of epigenetic modulators as possible treatments.

#### 2.1.1. Histone Modification

Posttranslational histone alterations, like DNA methylation, do not affect DNA sequencing but can alter its accessibility to transcriptional activity. However, histone alterations perform additional functions; the most well-known is their involvement in restoring DNA in the aftermath of cell injury [63]. There are numerous forms of histone alterations, with the most well-studied and significant regarding regulating the structure of chromatin and transcriptional function being acetylation, methylation, phosphorylation, and ubiquitination [64,65,66,67]. The alterations to histones are often mediated by specialized enzymes that operate at the histone N-terminal ends using amino acids such as lysine or arginine, as well as serine, threonine, tyrosine, and others [63,68,69].

**Histone Acetylation:** The equilibrium between HATs and HDACs regulates the acetylation of histone. Acetylation can diminish the positive charge present in lysine residues, which inhibits the attachment of histone ends to negatively charged DNA, exposing the underlying DNA [70]. Histone acetylation generally results in increased expression of genes. As a result, histone acetylation is commonly regarded as a functioning histone signal [71,72]. The equilibrium of histone acetylation and deacetylation is crucial in the control of cellular activities and growth. Histone acetylation instability has been linked to a variety of illnesses, involving cancer, neurological problems, and heart disease [73,74].

**Histone Methylation:** This normally happens via the addition of methyl groups to the lysine (K) residues of histone H3 and H4, representing one of the most essential post-transcriptional changes [75]. Histone methyltransferase (HMT) catalyzes this methylation by transferring methyl groups to histone lysine residues using S-adenosyl methionine (SAM) as the base [76]. Furthermore, acetylation and methylation at the identical lysine residues can function as opposites that hinder one another, resulting in histone interaction [77]. In HCC, histone methylation facilitates recruitment of Polycomb repressive complexes that silence TSGs such as CDKN2A and CDH1, promoting carcinogenesis [78]. Histone methylation instability has been increasingly implicated in the progression of HCC [79], where aberrant activity of histone methyltransferases (e.g., EZH2) and demethylases alters the expression of genes involved in tumor growth, angiogenesis, and immune evasion.

**Histone Phosphorylation:** It is among the most prevalent changes after translation, and also happens at histone protein serine as well as tyrosine sites [80]. Histone phosphorylation is involved in a variety of biological activities, such as the expression of genes, cell cycle control, DNA damage restoration, and asymmetric division of cells [81,82]. Histone3 phosphorylation at serine 10 (H3S10ph) is linked to transcriptional activation and is frequently seen near the promoter regions of actively transcribed genes. Phosphorylation at H3S10 facilitates recruitment of transcriptional coactivators, enhancing chromatin accessibility and gene activation [83,84]. Comparably, H3 at serine 28 (H3S28ph) is connected to gene expression. It stimulates the construction of transcriptional elongation complexes by promoting the induction of the BRD4 protein, which is a reader of acetylated histones [85]. H2A at serine 1 (H2AS1ph) and H2B at serine 10 (H2BS10ph), on the other hand, are related to gene suppression [86]. Various disorders, including cancer, have been linked to abnormal histone phosphorylation [87,88].

**Histone Ubiquitination:** Histone ubiquitination is a post-translational alteration in which one or more ubiquitin subunits are covalently attached to histone proteins. It is essential for regulating chromatin shape and gene activity [89]. H2A lysine 119 (H2AK119ub) and H2B lysine 120 (H2BK120ub) are two lysine sites located in the N-terminal ends of histone H2A and H2B that can be ubiquitinated. Histone ubiquitination is an important signaling pathway that affects a variety of chromatin-related functions [90,91]. It can influence the regulation of genes by modifying the access to chromatin and the induction of protein regulators. H2BK120ub, for instance, has been linked to transcriptional extension and gene expression stimulation [92]. It increases the binding of transcriptional activating elements that include the histone acetyltransferase p300/CBP and the FACT complex, which allows the enzyme RNA polymerase II to travel through the gene body [93,94,95]. H2AK119ub, on the other hand, is commonly related to transcriptional repression. It functions as a docking point for proteins with ubiquitin-binding domains, such as the Polycomb group (PcG) proteins, which are important in gene silencing and cellular identity preservation [96,97]. H2AK119ub promotes PcG unit attachment to certain genomic zones, resulting in the formation of restrictive chromatin domains [98]. Histone ubiquitination has also been linked to DNA damage reaction and restoration pathways. Histone ubiquitination imbalance has been linked to a variety of illnesses, notably cancer [99,100]. Figure 1 provides an integrated overview of the major epigenetic mechanisms contributing to hepatocarcinogenesis.

It is important to distinguish between the two major layers of epigenetic regulation that affect gene expression in HCC. On one hand, chromatin-modifying processes, such as histone acetylation, methylation, phosphorylation, and ubiquitination, directly influence DNA accessibility to the transcriptional machinery by altering chromatin structure. These modifications either promote or suppress transcription initiation [101]. On the other hand, ncRNAs regulate gene expression primarily at the post-transcriptional level. They exert their effects through mechanisms such as mRNA degradation, translational repression, or acting as competitive endogenous RNAs (ceRNAs) [102]. While both pathways are epigenetically driven, their points of action and regulatory mechanisms are distinct yet complementary in the context of hepatocarcinogenesis.

#### 2.1.2. Epigenetic Association of ncRNA in HCC

In recent years, there has been growing interest in understanding the role of ncRNAs in the epigenetic regulation of HCC. ncRNAs are a broad class of RNA transcripts that do not code for proteins but play important regulatory roles in a variety of biological operations [103]. Previously considered transcriptional noise, many ncRNAs are now recognized as essential regulators of gene expression through various epigenetic mechanisms. The next-generation sequencing technique was implemented to identify ncRNAs [104]. They are known to enhance normal cellular activities, but abnormal ncRNA is linked to cancer development and spread in several malignancies, including HCC [105,106,107,108,109,110]. In HCC, the transcriptionally produced ncRNA is epigenetically controlled by abnormally functioning DNA methylation and histone acetylation, which results in intrusive action. The sncRNAs and lncRNAs are two primary groups of ncRNAs involved in HCC [111,112,113,114].

The sncRNAs are <200 nucleotides long [115]. Small interfering RNA (siRNA), microRNA (miRNA), ribosomal RNA (rRNA), and transfer RNA (tRNA) are examples of sncRNAs [116]. The most well-studied sncRNA, however, is miRNA. MiRNAs are small RNA molecules with a length of 21–23 nucleotides [117]. They act as post-transcriptional moderators by adhering to complementary regions in messenger RNAs (mRNAs), causing them to degrade or suppress translations [118]. MiRNA instability has been seen in a variety of malignancies, especially HCC, in which they can function as tumor-causing genes or tumour suppressors [119]. Tumor suppressor miRNAs and onco-miRNAs have significance in the development of cancer [120]. Various miRNAs have been discovered as epigenetically controlled in HCC, which means that they are impacted by DNA methylation or histone changes [121]. MiR-375, for example, is typically downregulated in HCC, and its decreased expression is linked to regulator hypermethylation [122,123]. In addition to miRNAs, long non-coding RNAs (lncRNAs) have emerged as crucial epigenetic regulators in HCC.

LncRNAs are a diverse set of transcripts with >200 nucleotides but without protein-coding capabilities [124]. They have evolved as important participants in controlling genes, chromatin structure, and epigenetic alterations [125]. The aberrant regulation of lncRNAs has been linked to the emergence and growth of HCC [126,127]. There has been increasing indication that lncRNAs might communicate with chromatin-modifying structures and direct them to genomic loci, affecting the expression of gene patterns [125,128,129]. Numerous lncRNAs have been found to exhibit aberrant epigenetic regulation in HCC [130]. HOTAIR (HOX transcript antisense intergenic RNA), for instance, is increased in HCC and functions as a framework for the polycomb repressive complex 2 (PRC2), resulting in H3K27 trimethylation, and gene silencing [131,132]. In HCC, irregular histone acetylation at the initiators of lncRNA-LET, H19, and lncRNA-p21 is increased. Under hypoxic circumstances, lncRNA-LET activity was decreased by the acetylation of H3/H4 caused by HIF1-induced HDAC3 expression [133]. Furthermore, methylation influences the generation of lncRNA [134].

PIWI-interacting RNAs (piRNAs) represent a distinct class of small non-coding RNAs that interact with PIWI proteins to regulate gene expression, transposon silencing, and genomic stability. Although initially thought to function primarily in germline cells, piRNAs are now recognized for their roles in somatic tissues and cancer, including HCC [135]. Emerging evidence has shown that dysregulated piRNA expression is associated with hepatocarcinogenesis through mechanisms such as DNA methylation, chromatin remodeling, and post-transcriptional silencing. For instance, Liao et al. (2024) [136] demonstrated that pir-has-21691, a newly identified piRNA that promotes HCC progression by suppressing pyroptosis through the inhibition of the TLR4/NF-kB/NLRP3 cascade. It increases cell invasion, decreases tumor formation when deleted in vivo. These findings underscore the potential of piRNAs as novel epigenetic biomarkers and therapeutic targets in HCC management.

Circular RNAs (circRNAs) are a class of covalently closed-loop non-coding RNAs formed by back-splicing events. Unlike linear RNAs, circRNAs lack 5′ caps and 3′ tails, which grants them exceptional stability. A key function of circRNAs is their role as microRNA (miRNA) sponges, where they sequester miRNAs via complementary binding sites and thus relieve repression on target mRNAs. This miRNA sponging mechanism is central to the ceRNA hypothesis, which describes a complex post-transcriptional regulatory network where different RNA molecules, such as circRNAs, lncRNAs, and mRNAs, compete for shared miRNAs [137]. In hepatocellular carcinoma (HCC), dysregulated circRNAs and ceRNA networks contribute to oncogenic signaling, epithelial–mesenchymal transition (EMT), and drug resistance [138]. Their tissue-specific expression and stability make them promising biomarkers and therapeutic targets.

The improper functioning of ncRNAs in HCC entails changes regarding their epigenetic markings as well as changes in their expression rates. Epigenetic alterations, involving DNA methylation and histone modifications, can have a direct impact on ncRNA production in HCC [139,140]. For example, DNA methylation of some miRNA and lncRNA regulators can silence them, but histone changes can affect the availability of their regulating components [141,142]. Furthermore, ncRNAs can affect the epigenetic environment of HCC cells [143]. Some lncRNAs, for example, have been found to attract chromatin regulators to specific genomic areas, such as DNA methyltransferases and histone-modifying enzymes, therefore changing the epigenetic configuration of targeted genes [144,145].

Identifying the epigenetic connection of ncRNAs in HCC offers significant promise for improving early detection and personalized treatment approaches. Currently, diagnostic strategies such as serum alpha-fetoprotein (AFP) testing and imaging modalities like ultrasound and MRI are widely used, but they suffer from limited sensitivity and specificity, particularly in early-stage HCC [146]. In contrast, emerging epigenetic biomarkers, including circulating miRNAs like miR-21 and miR-122 [147], and lncRNAs such as HULC and UCA1 [148] have shown superior diagnostic performance in early detection, with enhanced specificity and potential for non-invasive liquid biopsy applications. The ncRNAs are now recognized as promising indicators for HCC diagnosis and therapeutic responsiveness [149]. Furthermore, addressing abnormal ncRNAs and the epigenetic alterations linked to them is growing as a possible treatment option for HCC regulating the expression or function of certain ncRNAs, as well as addressing the enzymes associated with their epigenetic control, has a chance to correct the abnormal patterns of gene expression seen in HCC [150,151,152]. Figure 2 shows the role of ncRNAs in HCC development.

Emerging evidence supports the role of ncRNAs as diagnostic and prognostic biomarkers in HCC. Circulating miRNAs such as miR-21 and lncRNAs like HULC have been proposed as non-invasive serum markers [2]. Moreover, therapeutic strategies targeting oncogenic ncRNAs or restoring tumor-suppressive ncRNAs hold promise in precision medicine approaches for liver cancer.

#### 2.1.3. DNA Methylation

DNA methylation markers include 5-methyl-cytosine (5mC), mediated by DNMT1, DNMT3A, and DNMT3B, and 5-hydroxymethylcytosine (5hmC), mediated by TET1, TET2, and TET3. DNA methylation controls gene expression. This epigenetic modification contributes to modulating gene expression and has emerged as a promising therapeutic target for early detection and prognostic assessment in HCC. Aberrant DNA methylation patterns are a hallmark of liver cancer progression. For instance, in a subset of liver cancer patients characterized by CD133^+^/CD44^+^ expression, the non-collagenous protein osteopontin (OPN) regulates DNA methylation and promotes metastatic progression. By suppressing DNMT1 expression and lowering the methylation of TSGs, such as GATA4, CDKL2, and RASSF1, OPN knockdown in CD133^+^/CD44^+^ inhibits cell migration [153]. A novel locus that is connected to the progression of HCC was created by fusing transcriptome data with epigenetic changes. These loci contain the COMT and FMO3 genes, which, when re-expressed in liver cancer cell lines, increase apoptosis and slow cell growth. These diversely expressed loci are also linked to immunological processes, including lymphocyte migration, and actively participate in the pathways. Differential DNA methylation patterns have been identified at specific loci in blood samples from patients with HCC. Interestingly, they found that hypomethylated genes such as *CSF2*, *IFITM5*, and *IL9* were associated with immunomodulatory processes during the pre-diagnostic phase of HCC. Thus, the study highlights the significance of epigenetic regulation in modulating genes implicated in HCC, particularly those serving as early diagnostic markers [154].

DNA hypomethylation primarily occurs in repetitive DNA sequences, or CpG sites, in liver cancer (apart from CpG islands). Genomic instability, frequent mutations, and rearrangements that take place in chromatin areas with inactive sites are the precursors to hypomethylation in DNA. It was discovered that global hypomethylation encouraged the production of transcription factors. According to research, the enhancer hypomethylation developed as a tool for cancer therapies to identify overexpressed transcription factors. A transcription factor called CEBP enhancer, which is overexpressed in patients with liver cancer, was shown to have hypomethylation throughout the entire genome after examination of affinity and bisulfite-based whole genome sequencing. These enhancers, which are non-coding, are linked to the detection of cells that are dysregulated and controlled by a combination of BRD4, H3K27ac, and transcription factors. However, it is a challenging procedure because cancer care options involve hypomethylating drugs. Additionally, it was shown that the promoter regions of the enzymes phosphoglycerate kinase (PGK1), pyruvate dehydrogenase kinase 1 (PDHK1), glycolytic, and protein kinase were considerably hypomethylated and phosphorylated. The pan-cancer research also revealed that higher PGK1 and PDHK1 mRNA levels are linked to advanced TNM stages and worse overall survival in several malignancies, including liver cancer. The majority of DNA hypermethylation occurs at cis-regulatory elements and promoter-associated CpG island sites (CGIs), which are linked to lower gene expression. The region-specific hypermethylation encourages tumour suppressor genes to be silenced. APC and CDKN2A are two hypermethylated gene promoters that can discriminate between the tumours of liver cancer and the surrounding nontumorous liver tissues [155,156,157,158].

TSGs are among the genes that are hypermethylated in liver cancer. TSGs such as *RASSF1*, *APC*, *CDKN1A*, *CDH1*, *NEFH34*, and *NOTCH3* regulate various molecular pathways involved in metastasis and tumor progression. These reveal dysregulated DNA methylation, which plays a significant role in the development of liver cancer and is essential for better survival rates. The biological functions of these TSGs vary depending on the milieu in which they are found, but further research is required to determine how they affect liver cancer. Therefore, DNA methylation is a crucial mechanism in TSG silencing. The increased activity of DNMT1 causes the hepatocyte growth factors (HGFs) in liver cancer to be hypermethylated. This promotes the hypermethylation of TSGs (LHX9, MYOCD, and PANX2) that are associated with tumour progression and metastasis. The development of liver cancer is significantly influenced by the epigenetic changes in the c-Met (HGF receptor). The increased expression of c-Met in liver cancer CTCs is caused by the decreased DNA methylation. Additionally, it has been demonstrated experimentally that the hypermethylation at the CGIs in liver cancer is connected to transcriptional activity [159,160,161].

The elevated expression of enzymes like DNMTs causes the modification in DNA methylation, which is connected to a poor prognosis. For instance, liver cancer upregulates the expression of DNMT3a and DNMT3b, while chronic hepatitis upregulates DNMT1 and DNMT3a. Thus, by encouraging survival pathways, these enzymes cause the growth of tumours. For example, the Snail pathway induces CpG methylation via DNMT1 expression at the promoter region of E-cadherin and supports dysregulated signalling cascades, including PI3K/Akt/GSK3 implicated in cancer growth and metastasis. Similar to this, in liver cancer, DNMT3 also facilitates the epigenetic activation of metastasis-associated protein 1 (MTA1). In HCC, DNMT3a-mediated methylation has been linked to upregulation of *MTA1*, a gene associated with EMT and metastasis. So, by methylating CRAF at arginine 100 and controlling RAS interaction with the MEK/ERK pathway, PRMT6 inhibits the growth of liver cancer. Therefore, a thorough understanding of epigenetic modifiers is crucial for the creation of tailored pharmacologic inhibitors for the treatment of liver cancer [162,163,164].

#### 2.1.4. mRNA Methylation

The epigenetic alteration of mRNA, known as m6A methylation, has a role in the carcinogenesis of HCC. Methyltransferase-like 3 (METTL3) and Wilms tumour 1-associated protein (WTAP) are components of the mRNA methyltransferase complex, which mediates the methylation process [165]. The progression of liver cancer is linked to the upregulation of METTL3 and WTAP. The epigenetic silencing of ETS1 caused by WTAP-mediated m6A alteration also affected the G2/M cell cycle via regulating the production of p21/p27. A tumour suppressor gene upstream of WTAP is ETS1. Similar to this, using a m6A-YTHDF2-dependent pathway, METTL3 epigenetically suppressed the suppressor of cytokine signalling 2 (SOCS2). It has been discovered that METTL3 targets the tumour suppressor SOCS2, and that this is accomplished by epigenetic m6A alteration. The stability of mRNA is impacted by the m6A reader YTHDF2. Thus, identifying these methylation alterations would help in developing effective treatment plans for liver cancer [166,167].

METTL3 overexpression has been correlated with poor prognosis, increased tumor grade, and reduced survival in HCC patients. High METTL3 activity enhances m6A modifications on tumor suppressor transcripts, leading to their degradation via YTHDF2 [168,169], thereby supporting oncogenic progression. Targeting METTL3 or its interaction with m6A readers represents a promising therapeutic strategy, and small-molecule inhibitors are currently under development. Additionally, m6A signatures are being explored for use in patient stratification and personalized therapy.

### 2.2. Bioinformatics Tools and Databases for Epigenetic Target Discovery

The advent of large-scale omics technologies has necessitated the use of computational tools to analyze epigenetic changes in HCC. Platforms like The Cancer Genome Atlas (TCGA) and the International Human Epigenome Consortium (IHEC) offer integrated genomic and epigenomic datasets across tumor types. Tools such as MethHC, UALCAN, and TIMER enable visualization and analysis of DNA methylation and histone modification patterns linked with gene expression and immune infiltration [170]. Meanwhile, ENCODE, GEO, and Roadmap Epigenomics provide functional annotation for regulatory regions and chromatin states in liver tissues [154]. Emerging integrative tools like EpiDISH and MethylMix help deconvolve cell-type-specific methylation signatures and correlate them with gene silencing events [171]. Leveraging these platforms can accelerate the identification of candidate epigenetic biomarkers and therapeutic targets in liver cancer.

## 3. Epigenetic Biomarkers

Epigenetic changes, such as DNA methylation, histone alterations, and ncRNAs, serve as biomarkers for the detection and prognosis of HCC. Several epigenetic changes are critical for the development and spread of HCC and work in concert to regulate EMT. For example, there is a negative correlation between the generation of E-cadherin and hypermethylation of the CDH1 transcriptional promoter region. Numerous EMT transcriptional variables control the epigenetically expressed CDH1 gene [172]. Additionally, Snail recruits histone demethylase lysine-specific demethylase 1 (LSD1) to increase demethylation of K4 on H3 (H2K4m2) to boost CDH1 transcriptional activation [173]. Snail’s elevated levels of transcription point to poor survival [174], making it crucial for directing treatment. The ncRNAs were also emphasized as new modulators of several genes involved in the development of illness. There is growing evidence that ceRNAs serve as indicators for many malignancies, including HCC. A meta-analysis also indicated that lncRNA is effective as a diagnostic marker following this [175]. As additional screening indicators for HCC, the Cancer Genome Atlas discovered 4 lncRNAs: LINC01093, RP11-486O12.2, RP11-273G15.2, and RP11-863K10.7 [176]. Most strongly related to GAS5 transcription in HCC are the overall survival rate and disease-free survival. Decreased production of GAS5 is associated with hepatic failure, and it serves as a unique non-Ig partner for BCL6 [177]. As a result, this offers a viable treatment approach for HCC therapy. Additionally, a receiver operating characteristic curve that has increased specificity and sensitivity was used to evaluate UCA1 blood levels to differentiate between HCC patients and healthy people. Increased serum UCA1 expressions were seen in HCC patients.

circRNAs exhibit high stability and hold significant promise as screening biomarkers for HCC. Comparing HCC to non-tumor tissue, prior investigations found that around 35 circRNAs had been reduced and 26 had been elevated [178]. hsa_circ_0005075 exhibited a variation concerning the overall dimension of the tumor and is associated with a bad prognosis among the 61 variably transcribed circRNAs [179].

Likewise, a circRNA microarray experiment revealed that HCC had variable expression of the genes hsa_circ_0128298 and hsa_circ_0004018 [180,181]. Research further identified 13,124 circRNAs linked to HBV liver tumors among patients by using cutting-edge bioinformatics technologies [182]. The findings from this research would thus serve as the starting point for further investigation into the molecular causes and treatment possibilities. Exosomal circRNAs produced by cells were also identified as biomarkers. In one case, exosomal circRNA_100284 was found in the serum of an individual who had been subjected to arsenite, which was associated with the individual’s prognosis [183].

Exosomal circPTGR1, which facilitates metastasis through the miR449a-MET process, has also been identified in the metastatic liver cancer cell line (LM3) [184]. Similar to this, circ-DB as well as adipose-derived exosomal circRNA decrease miR-34a, trigger the cyclin A2 signaling pathway, and encourage the development of HCC, and lessen damages to DNA. This discovery will help us comprehend how adipose tissue and the development of HCC are related. Additionally, miRNAs were marketed as biomarkers for the detection of HCC. As an example, a liquid biopsy of HCC collected using sera of individuals with HCC was the subject of a nested case–control study. Increased amounts of miR-133a, miR-192, miR-143, miR-29a, miR-145, miR-29c, as well as miR-505 were found in the findings [185]. Furthermore, miR-219-5p encourages metastasis in HCC by controlling CDH1 expression and being linked to invasion and a bad prognosis for the patient [186]. As an outcome, this becomes a prognostic indicator for HCC. The chemo-drug resistance of HCC cells is also determined by the biomarkers. In one instance, sorafenib-resistant HCC cells were shown to have erroneously functioning circRNAs [187]. In HepG2-S HCC cells as well as sorafenib-resistant Huh7-S HCC cells, RNA sequencing investigation employing CIRI (V2.0) software found 582 as well as 1717 circRNAs, respectively [188]. Overall, the correlation of EMT with epigenetic changes suggests new HCC diagnostic and therapeutic biomarkers. Future research is still necessary to create more sophisticated cancer metastasis indicators that utilize epigenetic modifiers, though Figure 3.

### miRNA Biomarkers in Diagnosis and Prognosis

MicroRNAs in HCC could be employed as biomarkers for therapy monitoring and diagnostics [189,190]. While conventional biomarkers like AFP lack sufficient predictive power in many patients, circulating miRNAs such as miR-483-5p have demonstrated diagnostic sensitivity and specificity values as high as 75.5% and 89.8%, respectively [191], suggesting a potential role in replacing or supplementing traditional diagnostic tools. MiRNA profiling at the genomic stage can be determined using technologies. Shen et al.’s [192] latest case–control epidemiological research in healthy individuals and those with HCC assessed the expression of circulating miRNAs linked to the HCC genome-wide. Among circulating biomarkers, miR-483-5p has emerged as a high-impact diagnostic candidate with 75.5% sensitivity and 89.8% specificity for distinguishing HCC from controls [192]. Other key miRNAs include miR-122, whose downregulation promotes HCC proliferation by impairing lipid metabolism and mitochondrial function, and miR-21, frequently overexpressed in HCC, known to activate the PI3K/AKT signaling pathway, supporting tumor growth and resistance [190]. Anwar et al. discovered the methylation of miRNA genes in certain cell lines, and putative miRNAs were subsequently utilised to evaluate human tissue specimens [190]. The findings revealed that only HCC patients exhibited hypermethylation in at least three miRNA genes. In 75 cancer samples, Chen et al. showed that methylation-mediated suppression of miR-129-2 led to an upregulation of oncogenic SOX-4 [193].

In one investigation, all HCC patients tested positive for HBV, which is consistent with previous studies identifying HBV infection as a major etiological factor in the pathogenesis of HCC [119]. Demethylating drugs and histone-deacetylating compounds are two epigenetic inhibitors that have shown promise for the treatment of HCC. The FDA in the United States has approved the use of four epigenetic antagonists, including DNMT and HDAC inhibitors. However, it is important to note that these drugs are currently approved for the treatment of hematological malignancies, and not yet for HCC. While several epigenetic therapies are being investigated in preclinical and early clinical trials for HCC (Table 2), none have received FDA approval specifically for this indication.

In a study where mRNA expression and DNA methylation were monitored in specimens from patients, the strategy of uniting two distinct kinds of biomarkers had been suggested for early HCC diagnosis. The researchers found that abnormal ABCB6 mRNA and DNA methylation rates were better predictors than a single marker by itself [194]. Ozen et al. suggested combining genetic along epigenetic indicators for HCC diagnostics and prognostics. Unlike many other cancers, HCC exhibits strong resistance to conventional chemotherapy and radiotherapy [195].

## 4. Conclusions

Epigenetic dysregulation in HCC is critical to its pathology, progression, and therapeutic resistance. This review highlights the importance of major epigenetic processes, including DNA methylation, histone modifications, and non-coding RNA regulation, not only in regulating gene expression but also as a potential prognostic and diagnostic biomarker. The elucidation of hypermethylated tumor suppressor genes, the deregulated histone marks, and abnormal ncRNAs can formulate a base understanding of hepatocarcinogenesis and give hope to therapeutic targets to unblock its occurrence. Nevertheless, there are a number of obstacles that are preventing the clinical applicability of epigenetic discoveries. First, HCCs are heterogeneous among patients and etiologies, which makes biomarker validation a challenge. Second, although several epigenetic modifications have been detected, very few have been functionally validated and used in clinical practice. Beyond that, the majority of the available epigenetic therapies are confined to hematological malignancies, and they have not yet shown significant efficacy in treating solid tumors, including HCC.

In the future, combining epigenomic profiling with other omics data (such as transcriptomics, proteomics, and metabolomics) has the potential to improve the discovery of biomarkers and patient stratification. Recent technological advances, including liquid biopsy and single-cell epigenomics, have the potential to provide non-invasive, real-time evolution tracking of tumors. Additionally, new epigenetic therapeutics like isoform-specific histone deacetylase inhibitors, non-coding RNA mimetics or antagonists, and epigenome editing with CRISPR will potentially address the shortcomings of conventional therapeutics by being more targeted, permanent, and patient-specific.

## Figures and Tables

**Figure 1 pharmaceuticals-18-01281-f001:**
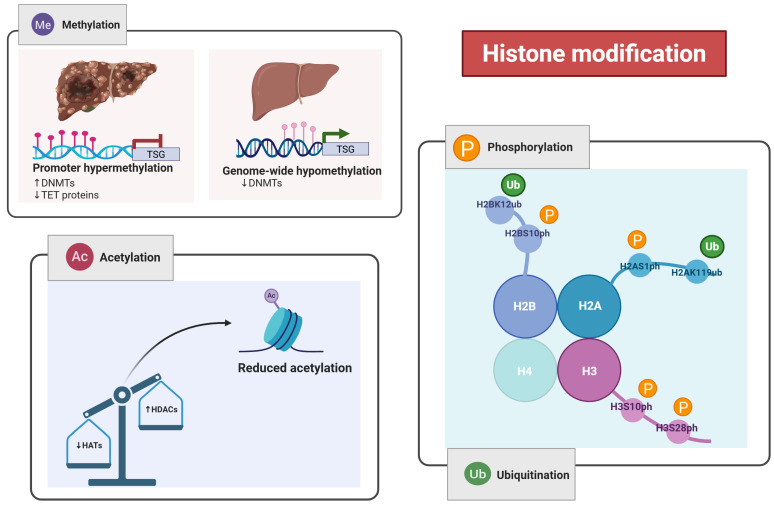
Integrated roles of major epigenetic mechanisms in hepatocellular carcinoma (HCC). This schematic highlights the interconnected pathways through which DNA methylation, histone modifications, and non-coding RNAs contribute to the regulation of gene expression in HCC. DNA methylation silences tumor suppressor genes, histone modifications influence chromatin accessibility, and non-coding RNAs mediate post-transcriptional gene regulation. Together, these mechanisms disrupt cellular homeostasis and promote tumorigenesis, angiogenesis, and metastasis in HCC. Created in BioRender. Babiker, R. (2025) https://BioRender.com/aoasc7p.

**Figure 2 pharmaceuticals-18-01281-f002:**
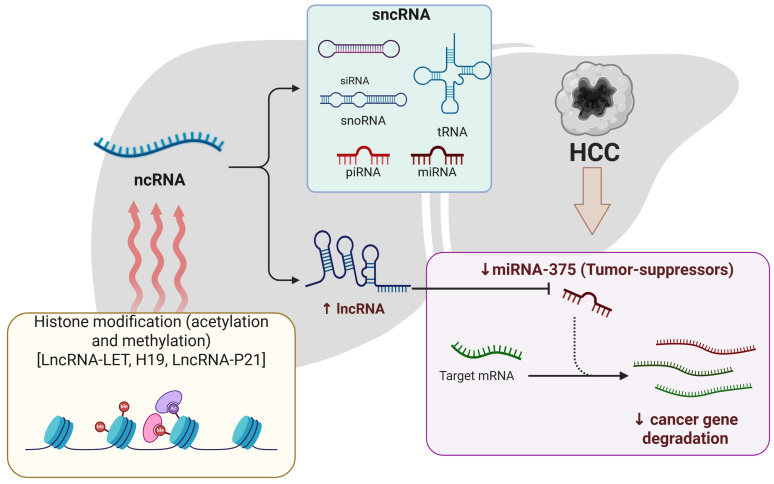
Illustrates the role of non-coding RNAs (ncRNAs) in hepatocellular carcinoma (HCC) development. Non-coding RNAs are divided into small non-coding RNAs (sncRNAs), including siRNA, snoRNA, piRNA, miRNA, and tRNA, and long non-coding RNAs (lncRNAs). Dysregulation of lncRNAs—driven by epigenetic modifications such as histone acetylation and methylation (e.g., LncRNA-LET, H19, and LncRNA-P21)—can suppress the expression of tumor-suppressive miRNAs like miRNA-375. This suppression leads to reduced degradation of oncogenic target mRNAs, promoting tumorigenesis and progression of HCC. Created in BioRender. Babiker, R. (2025) https://BioRender.com/0k3tinh.

**Figure 3 pharmaceuticals-18-01281-f003:**
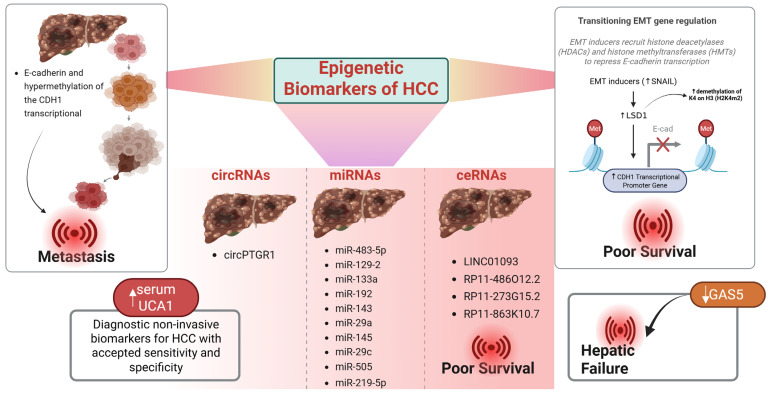
Epigenetic biomarkers contributing to the progression and prognosis of hepatocellular carcinoma (HCC). The dysregulation of various non-coding RNAs, including circular RNAs (circRNAs) like circPTGR1, microRNAs (miRNAs) such as miR-483-5p and miR-192, and competing endogenous RNAs (ceRNAs), including LINC01093 and RP11 variants, all of which are associated with poor survival outcomes. Epigenetic silencing of E-cadherin through CDH1 promoter hypermethylation and EMT-related histone modifications promotes metastasis. Upregulation of UCA1 in serum represents a non-invasive diagnostic biomarker for HCC with validated sensitivity and specificity. Additionally, downregulation of GAS5 is linked to hepatic failure, further underscoring the prognostic significance of epigenetic alterations in HCC. Created in BioRender. Babiker, R. (2025) https://BioRender.com/nuyjxx6.

**Table 1 pharmaceuticals-18-01281-t001:** Summary of Epigenetic Alterations and Their Roles in Hepatocellular Carcinoma (HCC).

Epigenetic Mechanism	Molecular Players	Functional Role in HCC	Biomarkers/Targets	Clinical Implication
DNA Methylation	DNMT1, DNMT3A, DNMT3B, TET1/2/3	Gene silencing, genomic instability	RASSF1, APC, CDKN2A, GATA4, CDKL2	Diagnostic and therapeutic value
Histone Modification	HATs, HDACs, HMTs, LSDs, PRMT6	Chromatin remodeling, DNA accessibility, gene regulation	H3K27me3, H3S10ph, H2AK119ub, H2BK120ub	Therapeutic value
Non-coding RNAs (ncRNAs)	miR-375, miR-219-5p, HOTAIR, GAS5, UCA1, lncRNA-LET	Regulation of transcription/translation, chromatin targeting, and metastasis	H19, circPTGR1, circ_0004018	Diagnostic and prognostic value
mRNA Methylation (m6A)	METTL3, WTAP, YTHDF2	Regulation of mRNA stability, cell cycle, suppressor gene silencing	SOCS2, ETS1	Therapeutic value
Epigenetic Biomarkers	miRNAs, lncRNAs, circRNAs	EMT, drug resistance, prognosis	miR-483-5p, LINC01093, has_circ_0005075	Diagnostic and prognostic value

DNMT1—DNA (Cytosine-5)-Methyltransferase 1; DNMT3A—DNA (Cytosine-5)-Methyltransferase 3 Alpha; DNMT3B—DNA (Cytosine-5)-Methyltransferase 3 Beta; TET1/2/3—Ten-Eleven Translocation Methylcytosine Dioxygenases 1, 2, and 3; RASSF1—Ras Association Domain Family Member 1; APC—Adenomatous Polyposis Coli; CDKN2A—Cyclin Dependent Kinase Inhibitor 2A; GATA4—GATA Binding Protein 4; CDKL2—Cyclin Dependent Kinase-Like 2; HATs—Histone Acetyltransferases; HDACs—Histone Deacetylases; HMTs—Histone Methyltransferases; LSDs—Lysine-Specific Demethylases; PRMT6—Protein Arginine Methyltransferase 6; H3K27me3—Trimethylation of Lysine 27 on Histone H3; H3S10ph—Phosphorylation of Serine 10 on Histone H3; H2AK119ub—Ubiquitination of Lysine 119 on Histone H2A; H2BK120ub—Ubiquitination of Lysine 120 on Histone H2B; m6A—N6-Methyladenosine; METTL3—Methyltransferase-Like 3; WTAP—Wilms Tumor 1-Associating Protein; YTHDF2—YTH N6-Methyladenosine RNA Binding Protein 2; SOCS2—Suppressor of Cytokine Signaling 2; ETS1—ETS Proto-Oncogene 1, Transcription Factor; EMT—Epithelial–Mesenchymal Transition.

**Table 2 pharmaceuticals-18-01281-t002:** Summary of current epigenetic therapies in HCC clinical/preclinical trials.

Drug Name	Target	Epigenetic Mechanism	Trial Phase	Combination Tested	Outcome Summary
Guadecitabine	DNMT1	DNA hypomethylation	Phase II	Sorafenib	Limited efficacy as monotherapy
Vorinostat	HDAC	Histone deacetylase inhibition	Phase I/II	Nivolumab	Enhances T-cell infiltration
Belinostat	HDAC	Histone acetylation	Phase I	Chemotherapy	Under investigation
SGI-110	DNMT	DNA demethylation	Preclinical	Checkpoint inhibitors	Shows potential synergy
Romidepsin	HDAC1/2	Histone deacetylation	Phase I	N/A	Toxicity limits standalone application

## Data Availability

No new data were created or analyzed in this study. Data sharing is not applicable to this article.

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
