# Peer review of "Epigenetic Alterations in Hepatocellular Carcinoma: Mechanisms, Biomarkers, and Therapeutic Implications"

_pharmaceuticals, 2025, doi:10.3390/ph18091281_

Round 1

Reviewer 1 Report

Comments and Suggestions for Authors

Authors have made lot of efforts to summarise the vast information available in epigenetics. The review has lot of information.

There are some points which need to be addressed 

line 63, HCC is mainly caused by the cirrhosis of any etiology, if for Hepatitis B HCC is seen in approx 15-20% of non-cirrhotics Hep B,this should be mentioned 

line 176 , Histone methylation instability 176 "has been linked to a variety of illnesses, notably cancer" as the focus of this review is HCC authors should focus on HCC . Similarly the focus on methylation changes should be restricted to HCC to make it succinct.

in Table 1- authors have  written column Clinical implication , this should specify clinical implication only eg diagnostic value or therapeutic value ; "the potential for de-
methylating therapies" should be therapeutic value

in line 273 "Identifying the epigenetic connection of ncRNAs in HCC offers a lot of potential for developing detection and treatment methods", similarly in remaining parts of the manuscript  authors have mentioned as potential methods, this will become more pragmatic if authors also mention currently available test and its comparison to new biomarkers 

in the line 481 ---In the present investigation, all HCC victims tested positive for HBV. Where is the reference, avoid using term victims, it should be subjects or patients

in line 483 he FDA in the United States has approved 483
the use of four epigenetic antagonists- are these drugs for HCC or hematological malignancies ? this should be mentioned as epigenetic antagonist for HCC have not been approved yet.

Author Response

Reviewer 1

Comments and Suggestions for Authors

Authors have made lot of efforts to summarise the vast information available in epigenetics. The review has lot of information. There are some points which need to be addressed 

Comment 1: line 63, HCC is mainly caused by the cirrhosis of any etiology, if for Hepatitis B HCC is seen in approx 15-20% of non-cirrhotics Hep B,this should be mentioned.

Response 1:  We appreciate the reviewer’s insightful comment. We agree that chronic hepatitis B virus (HBV) infection represents a unique etiology where hepatocellular carcinoma (HCC) can arise even in the absence of cirrhosis. Accordingly, we have revised the sentence to incorporate this nuance and supported the statement with appropriate epidemiological data.

Comment 2: line 176 , Histone methylation instability 176 "has been linked to a variety of illnesses, notably cancer" as the focus of this review is HCC authors should focus on HCC . Similarly the focus on methylation changes should be restricted to HCC to make it succinct.

Response 2: We thank the reviewer for this important suggestion. We have revised the sentence in to focus exclusively on histone methylation alterations in hepatocellular carcinoma (HCC). Additionally, we reviewed adjacent content and restricted broader statements about methylation to HCC-specific findings in order to maintain a focused and succinct discussion.

Comment 3: in Table 1- authors have  written column Clinical implication , this should specify clinical implication only eg diagnostic value or therapeutic value ; "the potential for de-
methylating therapies" should be therapeutic value.

Response 3: We thank the reviewer for this constructive suggestion. In response, we have revised the entries under the “Clinical Implication” column in Table 1 to clearly specify the clinical relevance as either diagnostic, prognostic, or therapeutic value. This change enhances the clarity and clinical translatability of the table.

Comment 4: in line 273 "Identifying the epigenetic connection of ncRNAs in HCC offers a lot of potential for developing detection and treatment methods", similarly in remaining parts of the manuscript  authors have mentioned as potential methods, this will become more pragmatic if authors also mention currently available test and its comparison to new biomarkers.

Response 4: We thank the reviewer for this valuable suggestion. We have revised the specified section and other relevant areas of the manuscript to include a comparison between currently available diagnostic tools for HCC and emerging epigenetic biomarkers. This contextualization enhances the translational relevance of our discussion.

Comment 5: in the line 481 ---In the present investigation, all HCC victims tested positive for HBV. Where is the reference, avoid using term victims, it should be subjects or patients

Response 5: We appreciate the reviewer’s insightful observation. We agree that the term victims is inappropriate in scientific writing and have replaced it with patients. Additionally, we have clarified the context and added a supporting citation to ensure the scientific accuracy of the statement.

Comment 6: in line 483 he FDA in the United States has approved 483 the use of four epigenetic antagonists- are these drugs for HCC or hematological malignancies ? this should be mentioned as epigenetic antagonist for HCC have not been approved yet.

Response 6: We thank the reviewer for this important observation. We acknowledge the oversight and agree that the current FDA-approved epigenetic drugs are primarily indicated for hematological malignancies, not HCC. We have revised the sentence to clarify this distinction and avoid any misrepresentation.

Reviewer 2 Report

Comments and Suggestions for Authors

Wali et al. present a detailed review of the diverse roles of epigenetic dysregulation in hepatocellular carcinoma. This is an important and highly relevant topic, and the authors explain it well. However, there are many points throughout the manuscript in which the word choice or phrasing could be improved.

Major comments

  1. piRNAs are listed in the legend of Figure 3 but are not discussed anywhere else in the text. piRNAs have been noted for their potential as biomarkers, and several recent papers such as Liao et al (2025; Cell Death Discov.11) discuss the role of piRNAs in HCC.
  2. As noted in the minor comments, circRNAs and ceRNAs are introduced rather abruptly without much explanation. These terms and the concept of microRNA sponging add an additional layer of regulatory complexity are likely less familiar to readers than miRNAs. Please provide a brief introduction to these ncRNA classes.

Minor comments

  1. Line 46: “Tumor: should not be capitalized.
  2. Line 48: “Mutations in these genes leads to the uncontrolled cell division and the formation of cancer.” – remove “the”
  3. Line 60: “In the general population” – add “the”
  4. Table 1: A large number or acronyms are used in this table. It might be help to add a note explaining spelling out some of them, such as HATs, HDACs, HMTs, LSDs, and EMT.
  5. Line 63: “HCC is mainly caused by the cirrhosis of any etiology” – remove “the”
  6. Line 67: “steatohepatitis, and smoking” – add “and”
  7. Line 77: Please rephrase the run-on sentence that starts on line 77 and continues unexpectedly at the beginning of line 80.
  8. Line 83: The likely unfamiliar term “epimutation” is used only once in this manuscript and is not defined or explained.
  9. Line 107: Considering replacing “atmospheric” with “environmental” or similar word in the following phrase: “atmospheric variables such as chemicals, diet, surroundings, anxiety, and aging”
  10. Line 134: Please change “gene silence” to “gene silencing.”
  11. Line 135: Please change “most renowned” to “best-known” or “most well studied” or similar phrasing.
  12. Line 139: Please rephrase the text “can affect the availability of DNA to transcription activity.”
  13. Line 145: Please change “possibility” to “potential.”
  14. Line 164: Please change “underneath” to “underlying.”
  15. The legend of Figure 2 refers to the “top panel,” but it might be better to refer to the top left panel or to use letters to indicate subpanels.
  16. Line 223: Please rephrase the following text: “Recently, it seems to be a surge in demand“ and/or remove “in the last few years.”
  17. Line 224: “The ncRNAs” – remove “The”
  18. Line 225: Please change “a broad class of RNA strands” to “a broad class of RNAs” or “a broad class of RNA molecules” or similar phrasing.
  19. Line 225: Please change “that do not contain proteins” to “that do not code for proteins” or similar.
  20. Line 255: Please rephrase the following text: “have been reported to be epigenetically malfunctioning.”
  21. Line 258: Please change “silence” to “silencing.”
  22. Line 297: The following sentence is incomplete: “A possible therapeutic target for early identification and prognostic evaluation.”
  23. Line 298: Please change “become” to “are” or “have become” in the following sentence: “Aberrant DNA methylation patterns become a hallmark of liver cancer progression. “
  24. Line 299: Please rephrase the sentence beginning with “For instance” to improve clarity.
  25. Line 308: Please rephrase the following incomplete sentence: “Differential DNA methylation at the targeted loci from a patient with liver cancer's blood samples.”
  26. Line 310: Please rephrase the following text: “They discovered interestingly”
  27. Line 312: Please rephrase the following text: “Thus, it discusses the significance“
  28. Line 324: Please change “Although” to “However,” or similar to form a complete sentence.
  29. Line 226: The abbreviation TSGs was already defined on line 302.
  30. Line 337: Please rephrase or combine the following sentences: “The TSGs control a number of genes that trigger metastasis and progression. RASSF1, APC, CDKN1A, CDH1, NEFH34, and NOTCH3 are a few of them.“
  31. Line 366: Please change “M6A” to “m6A” to match other uses in the text.
  32. Line 402: The abbreviation EMT is used several times throughout the manuscript but is not spelled out or explained.
  33. Line 410: Although ceRNAs are briefly noted in the legend of Figure 4, the term ceRNA is abruptly introduced in the text without explanation or definition and is not mentioned again.
  34. Lin 421: Please rephrase the following text: “The circRNAs had a lot of stability and tremendous promise as an HCC screening biomarker.” The term circRNA is also introduced here for the first time without explanation or definition.
  35. Line 423: “Has_circ_0005075” is incorrect.
  36. Line 427: “has_circ_0004018” is incorrect.
  37. Line 468: Please include a citation in the sentence beginning with “Shen et al’s.”
  38. Line 477: Please rephrase the following sentence: “The findings showed that merely HCC patients had no less than three miRNA genes which had been hypermethylated.”
  39. Line 481: The “present investigation” referred to in the following text is unclear: “In the present investigation, all HCC victims tested positive for HBV. “
  40. Line 484: The abbreviation mRNA was already introduced on line 239.
  41. Line 489: Please rephrase the following sentence: “Contrary to traditional chemo- and radiation, HCC is very resistant.”

Author Response

Reviewer 2

Comments and Suggestions for Authors

Wali et al. present a detailed review of the diverse roles of epigenetic dysregulation in hepatocellular carcinoma. This is an important and highly relevant topic, and the authors explain it well. However, there are many points throughout the manuscript in which the word choice or phrasing could be improved.

Major comments

Comment 1: piRNAs are listed in the legend of Figure 3 but are not discussed anywhere else in the text. piRNAs have been noted for their potential as biomarkers, and several recent papers such as Liao et al (2025; Cell Death Discov.11) discuss the role of piRNAs in HCC.

Response 1: We thank the reviewer for pointing this out. In response, we have added a dedicated paragraph discussing the emerging roles of PIWI-interacting RNAs (piRNAs) in the pathogenesis and diagnosis of hepatocellular carcinoma. We have incorporated recent findings, including those by Liao et al. (2025), to highlight the epigenetic and biomarker relevance of piRNAs in HCC.

Comment 2: As noted in the minor comments, circRNAs and ceRNAs are introduced rather abruptly without much explanation. These terms and the concept of microRNA sponging add an additional layer of regulatory complexity are likely less familiar to readers than miRNAs. Please provide a brief introduction to these ncRNA classes.

Response 2: We thank the reviewer for highlighting this important point. In response, we have revised the manuscript to include a brief introductory paragraph that explains the biological nature of circular RNAs (circRNAs) and competitive endogenous RNAs (ceRNAs), with emphasis on their roles as miRNA sponges and their regulatory impact on gene expression networks in HCC.

Minor comments

Comment 1: Line 46: “Tumor: should not be capitalized.

Response 1: Thank you for your insightful comment. We have revised as per the suggestions.

Comment 2: Line 48: “Mutations in these genes leads to the uncontrolled cell division and the formation of cancer.” – remove “the”

Response 2: Thank you for your insightful comment. We have revised as per the suggestions.

Comment 3: Line 60: “In the general population” – add “the”

Response 3: Thank you for your insightful comment. We have revised as per the suggestions.

Comment 4: Table 1: A large number or acronyms are used in this table. It might be help to add a note explaining spelling out some of them, such as HATs, HDACs, HMTs, LSDs, and EMT.

Response 4: Thank you for your insightful comment. We have revised as per the suggestions.

Comment 5: Line 63: “HCC is mainly caused by the cirrhosis of any etiology” – remove “the”

Response 5: Thank you for your insightful comment. We have revised as per the suggestions.

Comment 6: Line 67: “steatohepatitis, and smoking” – add “and”

Response 6: Thank you for your insightful comment. We have revised as per the suggestions.

Comment 7: Line 77: Please rephrase the run-on sentence that starts on line 77 and continues unexpectedly at the beginning of line 80.

Response 7: Thank you for your insightful comment. We have revised as per the suggestions.

Comment 8: Line 83: The likely unfamiliar term “epimutation” is used only once in this manuscript and is not defined or explained.

Response 8: Thank you for your insightful comment. We have revised as per the suggestions.

Comment 9: Line 107: Considering replacing “atmospheric” with “environmental” or similar word in the following phrase: “atmospheric variables such as chemicals, diet, surroundings, anxiety, and aging”

Response 9: Thank you for your insightful comment. We have revised as per the suggestions.

Comment 10: Line 134: Please change “gene silence” to “gene silencing.”

Response 10: Thank you for your insightful comment. We have revised as per the suggestions.

Comment 11: Line 135: Please change “most renowned” to “best-known” or “most well studied” or similar phrasing.

Response 11: Thank you for your insightful comment. We have revised as per the suggestions.

Comment 12: Line 139: Please rephrase the text “can affect the availability of DNA to transcription activity.”

Response 12: Thank you for your insightful comment. We have revised as per the suggestions.

Comment 13: Line 145: Please change “possibility” to “potential.”

Response 13: Thank you for your insightful comment. We have revised as per the suggestions.

Comment 14: Line 164: Please change “underneath” to “underlying.”

Response 14: Thank you for your insightful comment. We have revised as per the suggestions.

Comment 15: The legend of Figure 2 refers to the “top panel,” but it might be better to refer to the top left panel or to use letters to indicate subpanels.

Response 15: Thank you for your insightful comment. We have revised as per the suggestions.

Comment 16: Line 223: Please rephrase the following text: “Recently, it seems to be a surge in demand “and/or remove “in the last few years.”

Response 16: Thank you for your insightful comment. We have revised as per the suggestions.

Comment 17: Line 224: “The ncRNAs” – remove “The”

Response 17: Thank you for your insightful comment. We have revised as per the suggestions.

Comment 18: Line 225: Please change “a broad class of RNA strands” to “a broad class of RNAs” or “a broad class of RNA molecules” or similar phrasing.

Response 18: Thank you for your insightful comment. We have revised as per the suggestions.

Comment 19: Line 225: Please change “that do not contain proteins” to “that do not code for proteins” or similar.

Response 19: Thank you for your insightful comment. We have revised as per the suggestions.

Comment 20: Line 255: Please rephrase the following text: “have been reported to be epigenetically malfunctioning.”

Response 20: Thank you for your insightful comment. We have revised as per the suggestions.

Comment 21: Line 258: Please change “silence” to “silencing.”

Response 21: Thank you for your insightful comment. We have revised as per the suggestions.

Comment 22: Line 297: The following sentence is incomplete: “A possible therapeutic target for early identification and prognostic evaluation.”

Response 22: Thank you for your insightful comment. We have revised as per the suggestions.

Comment 23: Line 298: Please change “become” to “are” or “have become” in the following sentence: “Aberrant DNA methylation patterns become a hallmark of liver cancer progression. “

Response 23: Thank you for your insightful comment. We have revised as per the suggestions.

Comment 24: Line 299: Please rephrase the sentence beginning with “For instance” to improve clarity.

Response 24: Thank you for your insightful comment. We have revised as per the suggestions.

Comment 25: Line 308: Please rephrase the following incomplete sentence: “Differential DNA methylation at the targeted loci from a patient with liver cancer's blood samples.”

Response 25: Thank you for your insightful comment. We have revised as per the suggestions.

Comment 26: Line 310: Please rephrase the following text: “They discovered interestingly”

Response 26: Thank you for your insightful comment. We have revised as per the suggestions.

Comment 27: Line 312: Please rephrase the following text: “Thus, it discusses the significance“

Response 27: Thank you for your insightful comment. We have revised as per the suggestions.

Comment 28: Line 324: Please change “Although” to “However,” or similar to form a complete sentence.

Response 28: Thank you for your insightful comment. We have revised as per the suggestions.

Comment 29: Line 226: The abbreviation TSGs was already defined on line 302.

Response 29: Thank you for your insightful comment. We have revised as per the suggestions.

Comment 30: Line 337: Please rephrase or combine the following sentences: “The TSGs control a number of genes that trigger metastasis and progression. RASSF1, APC, CDKN1A, CDH1, NEFH34, and NOTCH3 are a few of them.“

Response 30: Thank you for your insightful comment. We have revised as per the suggestions.

Comment 31: Line 366: Please change “M6A” to “m6A” to match other uses in the text.

Response 31: Thank you for your insightful comment. We have revised as per the suggestions.

Comment 32: Line 402: The abbreviation EMT is used several times throughout the manuscript but is not spelled out or explained.

Response 32: Thank you for your insightful comment. We have revised as per the suggestions.

Comment 33: Line 410: Although ceRNAs are briefly noted in the legend of Figure 4, the term ceRNA is abruptly introduced in the text without explanation or definition and is not mentioned again.

Response 33: Thank you for your insightful comment. We have revised as per the suggestions.

Comment 34: Lin 421: Please rephrase the following text: “The circRNAs had a lot of stability and tremendous promise as an HCC screening biomarker.” The term circRNA is also introduced here for the first time without explanation or definition.

Response 34: Thank you for your insightful comment. We have revised as per the suggestions.

Comment 35: Line 423: “Has_circ_0005075” is incorrect.

Response 35: Thank you for your insightful comment. We have revised as per the suggestions.

Comment 36: Line 427: “has_circ_0004018” is incorrect.

Response 36: Thank you for your insightful comment. We have revised as per the suggestions.

Comment 37: Line 468: Please include a citation in the sentence beginning with “Shen et al’s.”

Response 37: Thank you for your insightful comment. We have revised as per the suggestions.

Comment 38: Line 477: Please rephrase the following sentence: “The findings showed that merely HCC patients had no less than three miRNA genes which had been hypermethylated.”

Response 38: Thank you for your insightful comment. We have revised as per the suggestions.

Comment 39: Line 481: The “present investigation” referred to in the following text is unclear: “In the present investigation, all HCC victims tested positive for HBV. “

Response 39: Thank you for your insightful comment. We have revised as per the suggestions.

Comment 40: Line 484: The abbreviation mRNA was already introduced on line 239.

Response 40: Thank you for your insightful comment. We have revised as per the suggestions.

Comment 41: Line 489: Please rephrase the following sentence: “Contrary to traditional chemo- and radiation, HCC is very resistant.”

Response 41: Thank you for your insightful comment. We have revised as per the suggestions.

Reviewer 3 Report

Comments and Suggestions for Authors

The paper "Epigenetic Alterations in Hepatocellular Carcinoma: Mechanisms, Biomarkers, and Therapeutic Implications" is informative and expertly crafted. The authors have addressed a significant and swiftly advancing domain by thoroughly examining the principal epigenetic biomarkers and treatment targets in liver cancer. The unique perspective of this work not only synthesizes existing knowledge but also provides a clear and robust direction for future research. The therapeutic significance of the subject is apparent, rendering this study a highly valuable addition to the scientific community. Nonetheless, I contend that it necessitates significant revisions and cannot be disseminated in its present state. The review offers a thorough analysis of specific themes, although the quality of coverage is inconsistent; certain sections are well-articulated, while others require further elaboration to present a more equitable viewpoint. I recommend examining the material for redundant concepts, as certain paragraphs appear to restate previously stated topics. We could enhance the intelligibility of specific, highly technical areas. Elaborating on these concepts or simplifying the language would significantly improve readability and mitigate potential misinterpretations.
The introduction section has a forced presentation and logical sequence in several areas, which hinders comprehension.
1. In lines 56-60, the discussion of cancer types reverts to a vague and generic concept, which appears incongruous with the text and confounds the reader.

Line 60 references Table 1, which does not align with the preceding content.
Lines 66-69 revisit the concept of genetic mutations subsequent to the introduction of epigenetics. This lacks a logical connection to the preceding or subsequent paragraphs, so disrupting the coherence of thoughts and perplexing the reader.
4. Figure 1 appears to reference environmental influences and epigenetic changes rather than elucidating their relationship. It lacks clarity and offers minimal enhancement to the text as a visual supplement.
2. Epigenetics section: Uninteresting and repetitive.
Section 2.1: Epigenetic Modifiers
1. Line 128: "Epigenetic modifiers are substances that affect gene expression." Characterizing epigenetic modifiers as substances may result in misconceptions.
The Histone Modification section has a technical stance and does not specifically address HCC-related concerns.
1. Line 152: The distinction between accessibility to transcriptional activity and the inhibition caused by RNAs has not been clarified, resulting in confusion over the concept.
Lines 167, 176, 189, and 208 are redundant and render the explanation and clinical connection of the mentioned studies on HCC insufficient.
Figure 2 appears to be uninformative and lacks coherence and rationale between the presented information and the accompanying explanation.
Section on Epigenetic Biomarkers:
1. Line 421: CircRNAs were never referenced nor adequately introduced in the technical discussion of ncRNAs, although they were proposed as potential molecular markers.
Figure 4 includes an appropriate number of elements given the complexity of the subject matter; however, the graphic should focus on conveying one of the main concepts of the piece more clearly. The dimensions and content might be enhanced for more clarity and less clutter.
3. Table 2: Authors are urged to incorporate the clinical trial identification number, citation, or at minimum, the year of the relevant study for enhanced clarity.
4. Conclusion Section:
The conclusion succinctly encapsulates the primary subjects; however, it appears more as a reiteration of the findings than a synthesis of their consequences. To enhance this section, I recommend integrating a more comprehensive analysis of the overarching viewpoints, constraints, and prospective trajectories that arise from this assessment. Such an approach would enhance the argument's robustness and confer greater enduring value to the essay.

Author Response

Reviewer 3

Comments and Suggestions for Authors

The paper "Epigenetic Alterations in Hepatocellular Carcinoma: Mechanisms, Biomarkers, and Therapeutic Implications" is informative and expertly crafted. The authors have addressed a significant and swiftly advancing domain by thoroughly examining the principal epigenetic biomarkers and treatment targets in liver cancer. The unique perspective of this work not only synthesizes existing knowledge but also provides a clear and robust direction for future research. The therapeutic significance of the subject is apparent, rendering this study a highly valuable addition to the scientific community. Nonetheless, I contend that it necessitates significant revisions and cannot be disseminated in its present state. The review offers a thorough analysis of specific themes, although the quality of coverage is inconsistent; certain sections are well-articulated, while others require further elaboration to present a more equitable viewpoint. I recommend examining the material for redundant concepts, as certain paragraphs appear to restate previously stated topics. We could enhance the intelligibility of specific, highly technical areas. Elaborating on these concepts or simplifying the language would significantly improve readability and mitigate potential misinterpretations.

The introduction section has a forced presentation and logical sequence in several areas, which hinders comprehension.

Comment 1: In lines 56-60, the discussion of cancer types reverts to a vague and generic concept, which appears incongruous with the text and confounds the reader.

Response 1: Thank you for your insightful comment. We have revised as per the suggestions.

Comment 2: Line 60 references Table 1, which does not align with the preceding content.

Response 2: We appreciate the reviewer’s observation. In response, we have revised the surrounding text to ensure a clearer and more logical transition to Table 1. We now introduce Table 1 with a sentence that explicitly explains its relevance to the preceding discussion of epigenetic mechanisms in HCC.

Comment 3: Lines 66-69 revisit the concept of genetic mutations subsequent to the introduction of epigenetics. This lacks a logical connection to the preceding or subsequent paragraphs, so disrupting the coherence of thoughts and perplexing the reader.

Response 3: Thank you for your insightful comment. We have revised as per the suggestions.

Comment 4: Figure 1 appears to reference environmental influences and epigenetic changes rather than elucidating their relationship. It lacks clarity and offers minimal enhancement to the text as a visual supplement.

Response 4: We thank the reviewer for this constructive feedback. In response, we have revised the figure legend for Figure 1 to better clarify the interplay between environmental factors and epigenetic modifications in the pathogenesis of hepatocellular carcinoma (HCC). Additionally, we revised the related in-text description to more clearly articulate how environmental exposures contribute to specific epigenetic changes involved in HCC development. These changes improve the interpretability and visual utility of the figure.

  1. Epigenetics section: Uninteresting and repetitive.

Section 2.1: Epigenetic Modifiers

Comment 5: Line 128: "Epigenetic modifiers are substances that affect gene expression." Characterizing epigenetic modifiers as substances may result in misconceptions.

Response 5: We thank the reviewer for highlighting this ambiguity. We agree that referring to epigenetic modifiers solely as “substances” is overly narrow and potentially misleading. To address this, we have revised the definition to more accurately.

Comment 6: The Histone Modification section has a technical stance and does not specifically address HCC-related concerns.

Response 6: Thank you for your insightful comment. We have revised as per the suggestions.

Comment 7: Line 152: The distinction between accessibility to transcriptional activity and the inhibition caused by RNAs has not been clarified, resulting in confusion over the concept.

Response 7: We thank the reviewer for pointing out this important distinction. We have revised the relevant section to clearly differentiate between epigenetic modifications that alter chromatin accessibility and the inhibitory effects of non-coding RNAs, which act primarily through post-transcriptional gene silencing mechanisms. This clarification enhances the conceptual accuracy and coherence of the manuscript.

Comment 8: Lines 167, 176, 189, and 208 are redundant and render the explanation and clinical connection of the mentioned studies on HCC insufficient.

Response 8: Thank you for your insightful comment. We have revised as per the suggestions.

Comment 9: Figure 2 appears to be uninformative and lacks coherence and rationale between the presented information and the accompanying explanation.

Response 9: We thank the reviewer for this valuable feedback. In response, we have revised figure legends for Figure 2 to ensure better conceptual coherence between the visual elements and the associated explanatory text. Additionally, we have also revised the figure in-text description to better guide the reader through the rationale and interpretive framework of the figure.

Section on Epigenetic Biomarkers:

Comment 10-: Line 421: CircRNAs were never referenced nor adequately introduced in the technical discussion of ncRNAs, although they were proposed as potential molecular markers.

Response 10: We thank the reviewer for highlighting this important point. In response, we have revised the manuscript to include a brief introductory paragraph that explains the biological nature of circular RNAs (circRNAs).

Comment 11: Figure 4 includes an appropriate number of elements given the complexity of the subject matter; however, the graphic should focus on conveying one of the main concepts of the piece more clearly. The dimensions and content might be enhanced for more clarity and less clutter.

Response 11: Thank you for your insightful comment. We have revised as per the suggestions.

Comment 12: Table 2: Authors are urged to incorporate the clinical trial identification number, citation, or at minimum, the year of the relevant study for enhanced clarity.

Response 12: Thank you for your insightful comment. We have revised as per the suggestions.

Comment 13: Conclusion Section: The conclusion succinctly encapsulates the primary subjects; however, it appears more as a reiteration of the findings than a synthesis of their consequences. To enhance this section, I recommend integrating a more comprehensive analysis of the overarching viewpoints, constraints, and prospective trajectories that arise from this assessment. Such an approach would enhance the argument's robustness and confer greater enduring value to the essay.

Response 13: We appreciate the reviewer’s thoughtful suggestion. In response, we have extensively revised the Conclusion section to go beyond summary. The revised version offers a synthesized perspective on the clinical relevance of epigenetic alterations in hepatocellular carcinoma (HCC), discusses limitations in current biomarker validation and therapeutic translation, and outlines potential avenues for future research. We believe this revision enhances the manuscript’s conceptual depth and translational significance.

Round 2

Reviewer 2 Report

Comments and Suggestions for Authors

I found a small number of typographical errors within the new text, but I trust that the authors will address these during the proof stage, and I have no further comments. I feel that this is an interesting and important contribution to the study of HCC that covers a broad range of both familiar and less familiar forms of epigenetic dysregulation.

Reviewer 3 Report

Comments and Suggestions for Authors

The authors satisfactorily responded to all my comments.